# Individual differences in personality predict the use and perceived effectiveness of essential oils

**Lindsay S. Ackerman, William J. Chopik** *

Department of Psychology, Michigan State University, East Lansing, Michigan, United States of America

* chopikwi@msu.edu

**Data Availability Statement:** Data, materials, and syntax are available at https://osf.io/r9c62/.

**Funding:** The authors received no specific funding for this work.

## Abstract

Essential oil (EO) use is growing in popularity and ostensibly used for treating or preventing various ailments or conditions. Despite the increase in use, there is a paucity of research on psychosocial predictors of EO use and their perceived effectiveness. However, several psychosocial characteristics are associated with health-promoting behavior and a tendency to believe in homeopathic cures. In the current study, we examined a variety of individual differences in the use and perceived effectiveness of essential oils in a sample of 1,202 participants ($M_{age}$ = 31.33, $SD$ = 13.77; 61.7% women, 75.6% Caucasian). We found that receptivity to pseudo-profound fabricated statements and religiosity were the most consistent predictors of greater use of, perceived effectiveness of, and a willingness to spend more money on EOs.

## Introduction

Essential oils (EOs) are "any of a class of volatile oils that give plants their characteristic odors and are used especially in perfumes and flavorings, and for aromatherapy" [1]. EOs are commonly used topically, internally, or by diffusion to prevent or treat illness or ailments, as substitutes for chemical-based cleaning products and candles, and for mood alteration or enhancement. EOs are used by many people; indeed, by some estimates, it was a $7.16 billion industry in 2017 [2] and is expected to increase to over $11 billion by 2022 [3]. However, there have been no studies to date examining the psychological predictors (i.e., personality) of EO use and effectiveness. In the current study, we examined individual differences in Big Five personality traits, bullshit receptivity (BSR), and need for cognition (NFC) as predictors of EO use and perceived effectiveness.

Advocates of EOs will point to the ostensible benefits they provide through their antioxidant, antibacterial, antifungal, antimicrobial, and antiplaque/antigingivitic properties, as well as their performance as an effective insect repellant [4–10]. In our review of the literature on EOs, many of these studies compare the effectiveness of EOs to existing treatments, placebos, no treatment, or in combination with existing treatments on varying outcome variables [e.g., hormone levels, brain activation, subjective evaluations of stress, chemical reactions with other substances; 11–21]. The purported benefits of EOs range from treating pain [in combination with conventional treatments; 22], to memory and mood enhancement [e.g., promoting

**Competing interests:** The authors have declared that no competing interests exist.

calmness, alertness, contentment; 23, 24], to insomnia relief [25, 26]. Although the vast majority of research purports the benefits of EOs, there is some research to suggest that certain oils *decrease* working memory performance and reaction time [27]. In the general public, and among food and drug regulators, there is often a sense of skepticism regarding the effectiveness of EOs. There are also few comprehensive reviews published providing evidence for the overall effectiveness for EOs. Particularly absent from the EO literature is a delineation of the exact biological mechanisms through which EOs provide benefits for people.

A separate but related question as to whether EOs provide marked health benefits is whether people *perceive* benefits of EO use. Indeed, there is a large literature on placebo effects, such that the belief that a medication provides benefits is associated with an increased odds of that medication actually working as intended [28, 29]. There are many other scenarios in which individuals' beliefs translate to better outcomes for them. Relatedly, many people benefit from broader belief systems more generally, such as those found in many organized religions. For example, people who consider themselves religious or spiritual—characteristics that do not always overlap [30, 31]—experience a number of health and well-being benefits compared to those who are not religious or spiritual, including increased longevity [32, 33]. There are many reasons for these benefits, including a connection to a broader community or power that may reduce stress and loneliness and the fact that they are more physically active—all factors which likely contribute to a healthier lifestyle [33–35]. In other words, religious and spiritual individuals engage in healthier behaviors and social activities as a result of the participation in their faith or connection to a higher power, even in the absence of proof of divine intervention on their health. Preventative or reparative health behaviors, such as using EOs, may be included in this broader umbrella—they may provide health benefits through some other byproduct of health monitoring. Thus, it is likely that the use of EOs might reflect a belief that they work and that they have preventative or restorative health properties.

Further, it is likely that some individuals are more likely to purchase and try EOs, which is not often the subject of EO research that relies on small placebo controlled and randomization studies to demonstrate their effectiveness. But what might predict the purchase, use, and perceived effectiveness of EOs? An individual's dispositional tendencies to act in certain ways (i.e., their personality) predict a number of outcomes related to health and financial decision making [36, 37]. Examining personality as a predictor of EO use seems like a logical next step.

One dominant model of personality is the Big Five personality framework [38]. These five broad, global traits—often referred to as the Big Five—are extraversion (traits like *outgoing* and *lively*), agreeableness (traits like *helpful* and *sympathetic*), neuroticism (traits like *moody* and *worrying*), conscientiousness (traits like *hardworking* and *responsible*), and openness to experience (traits like *imaginative* and *curious*). Personality is related to health and health-promoting behavior, of which EO use may be just one type of behavior. Research has found that conscientiousness predicts greater longevity [39], is negatively related to risky health behaviors, is positively related to behaviors that promote health [36, 40], and is associated with changes in health behavior across the lifespan [41]. High neuroticism has been found to predict attrition in exercise programs [42]. Personality has also been found to be related to pain relief from administration of a placebo treatment [43]. For instance, Peciña and colleagues [44] found that among the most predictive traits for placebo analgesia were agreeableness (positively) and neuroticism (negatively). Regarding the benefits of EO use, it could be possible that personality traits often associated with demonstrated placebo effects might also be associated with perceiving more benefits of EOs.

There are also several personal characteristics beyond the Big Five personality traits that might predict use and perceived effectiveness of EOs. For example, NFC (i.e., an inclination toward effortful cognitive activities) and BSR (i.e., endorsing information as profound/meaningful despite its vacuous nature) are two constructs that might also predict EO use and

perceived effectiveness. Both characteristics involve the (lack of) critical engagement in cognitive activities of skepticism. BSR, for instance, is related to the endorsement of conspiracy theories and paranormal belief, and is negatively related to measures of intelligence and cognitive reasoning [45]. Most pertinent to the current study is that people high in BSR are more likely to believe in the efficacy of common types of alternative medicines (e.g., homeopathy, energy healing). Some aspects of NFC are associated with more imaginative thinking as well, but sometimes in different directions. For example, many of the characteristics of NFC fall on the opposite end of a continuum than the endorsement of paranormal beliefs, magical ideation, and fantasies when researchers map out the psychometric coverage of the trait [46]. However, there is also some research to suggest individuals high on NFC might be *more* likely to believe in things like witchcraft and extraterrestrial life [47] or that NFC may be unrelated to paranormal beliefs entirely [48]. These traits (low NFC; high BSR) might also be associated with an endorsement of the benefits of medicinal products not endorsed by federal regulators or other reputable agencies (i.e., EOs).

In the current study, we took an exploratory approach and predicted EO use/effectiveness from the Big Five personality traits, NFC, and BSR. These variables were chosen based on their previous associations with health behavior [36]. NFC and BSR were chosen because we thought that individuals high on each trait might more or less likely to critically examine the potential benefits of medicinal products that are (often) not FDA approved. We tentatively hypothesized that agreeableness and BSR would be associated with greater use and perceived effectiveness of EOs. Based on the high health literacy of individuals high in conscientiousness and NFC [49–51], one could reasonably hypothesize that individuals high in both traits might be less likely to use EOs, but this hypothesis was not made a priori, and the analysis is therefore exploratory.

## Materials and method

### Participants

Participants were recruited from Amazon Mechanical Turk (MTurk; $n$ = 790) and an undergraduate subject pool ($n$ = 412). The samples were combined to maximize statistical power. We excluded an additional 210 participants because they indicated they didn't read the materials carefully. The final sample consisted of 1,202 participants ($M_{age}$ = 31.33, $SD$ = 13.77; 61.7% were women), including undergraduate students from Michigan State University and respondents to the study survey made accessible through Amazon Mechanical Turk (MTurk). Regarding race/ethnicity, 75.6% of the sample identified as white, 9.3% identified as black, 3.7% identified as Hispanic/Latino, and 11.3% identified as mixed race/other. MTurk participants were compensated $.50 for their participation in the survey. Undergraduate participants were compensated with course credit. No a priori decisions about sample size were made; we sought to collect as many participants as we could given our resources. Our sample size of 1,202 enabled us to find effects as small as $f^2$ = .008 (with 80% power and $\alpha$ = .05) or $f^2$ = .01 (with 95% power and $\alpha$ = .05).

The Michigan State Institutional Review Board approved this human subjects research (IRB# 16-1291e). Participants consented electronically and data were analyzed in an anonymous way. Data, materials, and syntax are available at https://osf.io/r9c62/.

### Measures

**Personality.** Personality was measured using the Big Five Inventory-2 [BFI-2; 52]. The questionnaire contains 60 items that ask individuals to rate the extent to which each statement accurately describes them (i.e., "I am someone who . . .") on a scale ranging from 1(*disagree strongly*) to 5(*agree strongly*). The BFI-2 measures the personality domains of extraversion (12 items; sample item: "Is outgoing, sociable;" $\alpha$ = .83), agreeableness (12 items; sample item: "Is

helpful and unselfish with others;" $\alpha$ = .84), conscientiousness (12 items; sample item: "Is efficient, gets things done;" $\alpha$ = .88), neuroticism (12 items; sample item: "Is moody, has up and down mood swings;" $\alpha$ = .89), and open-mindedness (i.e., openness to experience; 12 items; sample item: "Is curious about many different things;" $\alpha$ = .87). Responses were averaged to create composites for each dimension.

**Bullshit receptivity.** BSR, as measured by reception of pseudo-profound fabricated statements, was assessed using the 30-item Bullshit Receptivity Scale [45]. Participants were given a series of statements that retained grammatical syntax of a sentence but were devoid of any true meaning and rated them on a five-point scale ranging from 1(*not at all profound)* to 5(*very profound*). Examples of these nonsensical statements include, *"Imagination is inside exponential space time events"*, and *"As you self-actualize, you will enter into infinite empathy that transcends understanding."* Responses were averaged to create an overall scale of BSR ($\alpha$ = .97).

**Need for cognition.** NFC, or an individual's tendency to prefer effortful cognitive activities, was assessed using a 6-item Need for Cognition scale [NCS-6; 53, 54]. Participants rated the items such as *"I only think as hard as I have to"* and *"The notion of thinking abstractly is appealing to me"* on a 5-point scale ranging from 1(*not at all like me*) to 5(*very much like me*). Responses were averaged to create an overall scale of NFC ($\alpha$ = .80).

**Essential oils use and effectiveness questionnaire.** EO use and perception of effectiveness were measured using a questionnaire developed specifically for this study. This questionnaire first asked questions about EO use (e.g., "do you currently use essential oils?") and method of application (topical, internal, diffusion), and then solicited reasons for using EOs. Reasons for currently using EOs included: (1) to provide relief for physical ailments/conditions, (2) to alter mental/emotional state, (3) to avoid physical/mental illness, (4) to improve mood, (5) to help sleep, (6) to enhance spiritual life, (7) to improve relationships with family or friends, (8) as a dietary supplement, or (9) to clean/disinfect. These questions were only asked among people who were currently using essential oils and participants could nominate multiple reasons for using EOs (e.g., for relieving physical ailments/conditions *and* enhancing their spiritual life).

If participants responded positively to using EOs for each of these purposes, they were then asked if they experienced the benefits they were seeking on a five-point scale ranging from 1 (*always*) to 5(*never*). In other words, if someone noted that they used EOs to enhance their spiritual life, they were then asked if they experienced the spiritual effects they were seeking. Responses to the perceived effectiveness of each reason was reverse scored so that higher values reflected more effectiveness and analyzed separately.

Finally, participants were asked about brands of EOs they used, how much money they'd be willing to spend on EOs per month (on an 8-point scale ranging from 1[*less than $10*] to 8 [*more than $200*]), and overall perceived effectiveness (i.e., "Overall, do you think essential oils are effective?") ranging on a scale from 1(*always*) to 5(*never*), which was recoded so that higher values indicated more effectiveness.

**Control variables.** Age, gender, household income (1[*less than $5,000*] to 9[*$100,000 or greater*]), religiosity (1[*not at all religious*] to 7[*extremely religious*]), and political orientation (1[*very liberal*] to 7[*very conservative*]) were all entered as control variables in logistic and linear regression analyses.

## Results

Overall, 66% of the sample currently used EOs. The most common brand of EOs used were NOW Essential Oils (27.5%), Edens Garden (22.6%), and doTerra (20.3%). Participants were willing to spend on average between $10 and $49 per month on essential oils ($M$ = 2.14, $SD$ = 1.22 on the 8-point scale). Among EO users, the overall effectiveness was slightly above

the midpoint on a 5-point scale ($M$ = 3.12, $SD$ = 1.07; $t$ = 3.25, $p$ < .001 compared to the mid-point), suggesting that users generally thought EOs were effective.

In the current sample, 25% ($n$ = 296) of people had previously tried EOs and stopped using them. The most common reasons for discontinuing use was that they were too expensive (38.9%) and that they did not experience the claimed benefits of EOs (32.2%). We restricted our analyses below to individuals currently using EOs and those who have never used EOs because we thought it was a more interesting comparison to examine people who have either had positive (currently using) or no experiences with EOs, rather than individuals who likely have a negative impression of EOs. In predicting who these former EO users are (with respect to personality and demographic characteristics), former EOs users were less conscientious (Exp($b$) = .77, $p$ = .05), more open to experience (Exp($b$) = 1.37, $p$ = .03), less receptive to bullshit (Exp($b$) = .67, $p$ < .001), and more likely to be men (Exp($b$) = .70, $p$ < .001). Some individuals reported stopping use because they did not experience any benefits; none of the personality and demographic characteristics significantly predicting halting use because of not receiving benefits ($ps$ > .07).

Logistic (for use: 0 = not use; 1 = use) or linear regressions were used to predict each outcome from the Big Five personality traits, BSR, and NFC while controlling for age, gender, income, religiosity, and political orientation. Full regression tables can be found in S1 through S25 Tables. The results are summarized in Table 1.

The largest and most consistent predictor of essential oil use and perceived effectiveness was BSR–it positively predicted all applications of use except for by diffusion, every reason for use, and nearly every perceived effectiveness of the various uses, as well as willingness to spend money and perceived overall effectiveness. In contrast to our predictions, the more agreeable someone was, the less likely they were to use EOs for a variety of reasons (although they are more likely to use EOs by diffusion). Neuroticism predicted a higher likelihood of use of oils topically or by diffusion, and to improve mental state. Those high in neuroticism were less likely to use EOs to improve social relationships, for dietary reasons, or to enhance their spiritual life. Openness to experience negatively predicted current use or internal administration, as well as use to avoid mental/physical problems, improve sleep and social relationships, or for dietary purposes. Those high in conscientiousness were less likely to use EOs to for various reasons. Extraversion and NFC were rarely predictors of use.

Interestingly, religiosity as a predictor of EO use and perceived effectiveness was also found to be significant for most variables measured (see supplementary tables). In fact, high religiosity positively predicted every use of EOs except for by diffusion and to improve mood or help sleep. Religiosity also positively predicted perceived effectiveness for relieving physical ailments, altering mental/emotional state, helping with sleep, enhancing spiritual life, and sustaining/improving relationships. Additionally, those high in religiosity were more likely to spend money on EOs and to find EOs effective overall.

Of the remaining control variables, women were more likely to be currently using EOs, to use them topically or by diffusion, and to use them for physical ailments, sleep, and as a dietary supplement. They were also more likely to spend money on EOs; however, they did not rate EOs as effective for use with any particular problem. Age positively correlated with EO use by diffusion and for sleep and cleaning. Older adults deemed EOs to be less effective for relieving sleep problems and as a dietary supplement. Political orientation (i.e., being conservative) was found to be positively predictive of topical use and use for spiritual enhancement. Those who were more conservative thought EOs were effective for relieving physical ailments, for avoiding physical/mental illness, for spiritual enhancement, and to sustain/improve relationships. Income was least likely to predict EO use but did positively predict topical application and use for spiritual enhancement. Income also positively predicted willingness to spend money on EOs.

**Table 1. Personality predicting essential oil use.**

|  | Extraversion | Agreeableness | Conscientiousness | Neuroticism | Openness | Bullshit Receptivity | Need for Cognition |
|---|---|---|---|---|---|---|---|
| Currently Use (Exp(b)) | 1.24 | .90 | 1.35 | 1.08 | **.68** | **1.70** | 1.00 |
| Topical Use (Now) (Exp(b)) | **1.54** | **.72** | 1.11 | **1.30** | 1.00 | **1.24** | .95 |
| Internal Use (Now) (Exp(b)) | 1.18 | .68 | .66 | 1.17 | **.43** | **1.64** | **1.59** |
| Diffusion Use (Now) (Exp(b)) | **1.46** | **1.36** | 1.10 | **1.46** | 1.35 | .88 | 1.08 |
| Why did you use? |  |  |  |  |  |  |  |
| Physical Ailment (Exp(b)) | 1.08 | .94 | .91 | 1.12 | .97 | **1.35** | .94 |
| Effectiveness (β) | .04 | .08 | -.06 | .02 | **-.17** | **.21** | .02 |
| Mental State (Exp(b)) | 1.33 | 1.09 | .76 | **1.42** | 1.05 | **1.37** | .89 |
| Effectiveness (β) | .07 | -.03 | -.04 | -.06 | -.05 | **.18** | .03 |
| Avoid Phys/Men Probs Exp(b)) | 1.33 | **.65** | **.62** | .87 | **.54** | **1.96** | 1.15 |
| Effectiveness (β) | .03 | -.01 | .06 | -.04 | -.15 | **.19** | .11 |
| Improve Mood (Exp(b)) | 1.07 | 1.00 | .80 | 1.21 | 1.13 | **1.61** | .91 |
| Effectiveness (β) | .04 | -.03 | .10 | **-.18** | -.06 | **.18** | .02 |
| Improve Sleep (Exp(b)) | 1.32 | 1.26 | **.64** | 1.08 | **.70** | **1.47** | 1.16 |
| Effectiveness (β) | **.14** | .02 | -.03 | -.003 | .01 | **.13** | -.08 |
| Improve Social Relations (Exp(b)) | .96 | **.45** | **.40** | **.52** | **.41** | **5.10** | 1.19 |
| Effectiveness (β) | .08 | **-.26** | -.04 | -.01 | .14 | .05 | -.05 |
| Dietary (Exp(b)) | 1.08 | **.48** | **.47** | **.63** | **.36** | **2.94** | **1.64** |
| Effectiveness (β) | -.07 | -.12 | .02 | **.16** | .03 | **.30** | .02 |
| Clean/Disinfect (Exp(b)) | .86 | **.72** | .77 | .94 | .77 | **1.74** | 1.32 |
| Effectiveness (β) | -.03 | .05 | .02 | -.01 | .02 | .12 | .11 |
| Spirituality (Exp(b)) | .87 | **.48** | **.53** | **.60** | .93 | **2.88** | .84 |
| Effectiveness (β) | -.19 | -.14 | .05 | -.08 | .07 | .14 | .17 |
| Main Outcomes |  |  |  |  |  |  |  |
| Willingness to Spend Money (β) | .02 | -.08 | -.04 | -.03 | **-.09** | **.26** | -.02 |
| Overall Effectiveness (β) | .05 | .04 | -.01 | .03 | -.03 | **.28** | -.03 |

Bolded coefficients are significant at p < .05.

## Discussion

In the present study, our most consistent finding was that those high in BSR were 70% more likely to use essential oils and were more likely to find them effective. Personality characteristics identified in previous research showing associations with similar phenomena (e.g., placebo effects, health behavior, homeopathic medicine) were largely unrelated to EO use. Investigation into the predictors of EO use is useful for many fields—to psychologists, medical professionals, and industry leaders alike—as the EO market is a massive and growing industry. Discovering who uses EOs and for what purposes can provide additional insight to practitioners helping clients make informed health decisions.

There is often uncertainty about the use and effectiveness of EOs, and very few studies provide evidence for EO's overall effectiveness. Even so, EOs remain a widely popular remedy for various ailments and issues. Until now, no research has investigated what might predict EO use in the general public, nor the traits of those who experience their benefits. Religious individuals and those high in BSR were more likely use EO for a variety of reasons and were more likely to find their use to be effective.

The fact that religious individuals were more likely to use EOs is consistent with other studies showing that they are more likely to engage in preventative and restorative health behaviors

—which contribute to some of the reasons why they live longer [33]. Religious individuals have better biological health, longevity, physical activity, and well-being [55]. Relevant to the current study, there is also some evidence that spiritual individuals are more likely to use some forms of traditional health care (e.g., doctors' appointments, methods to extend life)[56]. In a way, our finding that religious individuals are more likely to use EOs similarly captures the effect that religious individuals may use a variety of methods—including those that involve some faith beyond reason—to enhance their health. Worth noting, oil has a unique place in many religious traditions as a way of communicating hospitality, providing an incarnation of divine blessing, or as a form of medicine [57]. Although the modern, commercial use of EOs may not have religious origins, it is similar to historical and religious uses of oil in that religious people (and others) might be putting faith in EOs to cure illnesses or alleviate their negative life circumstances, as they do with other forms of complementary and alternative medicine [58].

In their original conceptualization of BSR, Pennycook and colleagues [45] noted that there are several definitions of bullshit, ranging from simple "nonsense" to something that is designed to impress but was constructed without concern for the truth [59]. In evaluating the appeal of EOs, we acknowledge that portions of this definition of bullshit may apply. For example, EOs purport to alleviate an impressive number of health and life problems despite there being no consensus on the effectiveness of these EOs for serving those purposes. In this way, EOs are designed to impress with their effectiveness despite their development and testing being constructed without a high standard for truth (e.g., well-powered randomized control trials with an active control, FDA oversight). People who find meaning in otherwise meaningless stimuli (i.e., those high in BSR) therefore are more likely to uncritically accept EOs' purported benefits at face value given that the mechanisms leading to the benefits are unknown or illusory. Those low in BSR may find such claims of EOs as lacking meaning and therefore unconvincing in their decisions to deal with a health or life issue. Because essential oils are likely harmless (but possibly not beneficial either), people may be less likely to scrutinize the claims about their effectiveness [60]. Of course, our paper is not meant to adjudicate the evidence about the effectiveness of EOs; rather we sought to examine the individual difference characteristics associated with EO use and perceived effectiveness. People high in BSR were more likely to use EOs and perceive their benefits, suggesting that they might also be susceptible to the claims of the benefits of EOs—particularly those that are most questionable regarding the underlying mechanisms (e.g., spiritual reasons).

The results from the current study create some exciting opportunities for future research. Are people high in BSR more receptive to placebo effects? Only well-powered and controlled randomized control trials that also measure BSR can answer this question. Another research question posed by the current study is whether EOs tend to be used in combination with (or instead of) more scientifically based methods for improving life problems (e.g., seeking therapy, going to a physician, engaging in preventative health behaviors [diet, exercise]). Are specific people, to their detriment, using EOs in place of more proven treatments? If they are using EOs instead, are they working as people say there are? Finally, future research could further examine the individuals who reported having used EOs in the past, but had stopped; specifically, understanding the traits of those who stopped because they believed the EOs were not effective would be of great interest.

There are some limitations of the current study that are worth acknowledging. For example, there was a substantial part of the sample (25%) who had tried—but ultimately ceased—using EOs. Unfortunately, because we were focused on the perceived effectiveness of current EO use, we missed an opportunity to ask more questions about people who stopped using EOs. In the future, researchers should study these individuals more in depth. The survey was also cross-sectional and relied on people's report of their typical everyday use of EOs. Future research

could more precisely measure EO use and even randomly assign participants to use EOs (or a placebo) in order to measure their effectiveness (particularly among those high in BSR). Alternatively, longitudinal studies of the health of EOs users over time would also be useful to conduct.

An additional limitation that must be acknowledged relates to the measurement of the constructs examined in the current study. For example, we assessed religiosity with a single-item measure. Although this was done for expediency reasons, both religiosity and spirituality (which we did not measure in this study) are separable, multi-faceted phenomena that can be better measured using more detailed scales [30]. For example, some conceptualizations of religiosity highlight different facets of religiosity [61]—an engagement in more structural/organized elements (i.e., external), personal identification and living a principled life (i.e., intrinsic), and an engagement with existential questions about life more broadly (i.e., quest). Future research can include such nuanced measures and a formal, established measure of spirituality to examine their utility in predicting the likelihood of using complementary and alternative medicines. Further, the use of the BSR measure is not without controversy and has been criticized on the grounds of both conceptual framing and cultural-specificity [62]. Specifically, Dalton takes issue with the BSR measure because there are scenarios in which some of the statements, despite being randomly generated, indeed make sense to practitioners of Eastern religions and may tap into an individual's propensity of wise reasoning. In a succinct critique, he offers that, "beauty, like bullshit, may be in the eye of the beholder" (p. 122) and that, just because someone from the West cannot immediately discern wisdom from a statement, does not necessary imply that a statement is bullshit. We take Dalton's critique to heart [although see Pennycook et al.'s response; 63]. It was not our intention to label EO users as being full of bullshit, bullshitters, or as naïve, uncritical consumers of medicinal products. Nor are we trying to discredit EOs or complementary and alternative medicines more broadly. Rather, we took the opportunity here to examine a variety of individual difference predictors of EO use, perceived effectiveness, and consumption. We found that this particular, albeit narrow, conceptualization of BSR was the most consistent predictor of each of these outcomes. Future research can more accurately determine why BSR is related to EO use and whether there is some overlapping variance with other constructs that might explain why some parts of BSR are associated with EO use [64].

Further, in the current sample, few people used EOs to enhance spiritual life (N = 253), to sustain/improve relationships (N = 195), or as a dietary supplement (N = 198), and thus these analyses had smaller sample sizes. As such, even though many predictors of these uses were statistically significant, the findings should be interpreted with caution given the small sample sizes. Lastly, there is significant variability in types of EOs, and brands of EOs may differ in quality or safety. The present study considered all EOs on a level playing field, yet perhaps there are certain types of oils (e.g. peppermint, lavender), certain brands (e.g. DoTerra, Eden's Garden), or certain methods of administration that are perceived as more effective.

In sum, with the growing prevalence of EO use and the lack of research on the effectiveness of EOs, the current study set out to examine reliable psychological predictors of EO use and perceived effectiveness. BSR and religiosity were the most consistent predictors; those high in these traits were more likely to use oils and find them effective in almost every instance. Future research can more directly examine *why* certain people report benefits while others do not.

## Supporting information

**S1 Table. Models predicting whether people ever use essential oils.**
(DOCX)

**S2 Table. Models predicting whether people currently use essential oils.**
(DOCX)

**S3 Table. Models predicting whether people currently use essential oils topically.**
(DOCX)

**S4 Table. Models predicting whether people currently use essential oils internally.**
(DOCX)

**S5 Table. Models predicting whether people currently use essential oils by diffusion.**
(DOCX)

**S6 Table. Models predicting whether people currently use essential oils for physical ail-
ments.**
(DOCX)

**S7 Table. Models predicting whether people currently use essential oils to alter mental/
emotional state.**
(DOCX)

**S8 Table. Models predicting whether people currently use essential oils for avoiding physi-
cal/mental illness.**
(DOCX)

**S9 Table. Models predicting whether people currently use essential oils to improve mood.**
(DOCX)

**S10 Table. Models predicting whether people currently use essential oils to help sleep.**
(DOCX)

**S11 Table. Models predicting whether people currently use essential oils to enhance spiri-
tual life.**
(DOCX)

**S12 Table. Models predicting whether people currently use essential oils to improve/sus-
tain relationships.**
(DOCX)

**S13 Table. Models predicting whether people currently use essential oils as dietary supple-
ment.**
(DOCX)

**S14 Table. Models predicting whether people currently use essential oils to clean/disinfect.**
(DOCX)

**S15 Table. Models predicting the effectiveness of EO for relieving physical ailments.**
(DOCX)

**S16 Table. Models predicting the effectiveness of EO for altering mental/emotional state.**
(DOCX)

**S17 Table. Models predicting the effectiveness of EO for avoiding physical/mental illness.**
(DOCX)

**S18 Table. Models predicting the effectiveness of EO to improve mood.**
(DOCX)

**S19 Table. Models predicting the effectiveness of EO to help sleep.**
(DOCX)

**S20 Table. Models predicting the effectiveness of EO to enhance spiritual life.**
(DOCX)

**S21 Table. Models predicting the effectiveness of EO to sustain/improve relationships.**
(DOCX)

**S22 Table. Models predicting the effectiveness of EO to clean/disinfect.**
(DOCX)

**S23 Table. Models predicting the effectiveness of EO as a dietary supplement.**
(DOCX)

**S24 Table. Models predicting willingness to spend money on EO.**
(DOCX)

**S25 Table. Models predicting overall effectiveness of EO.**
(DOCX)

## Acknowledgments

We would like to thank Sam Warshaw for his comments on a previous version of this manuscript.

## Author Contributions

**Conceptualization:** Lindsay S. Ackerman, William J. Chopik.

**Data curation:** William J. Chopik.

**Formal analysis:** William J. Chopik.

**Investigation:** Lindsay S. Ackerman, William J. Chopik.

**Methodology:** William J. Chopik.

**Project administration:** William J. Chopik.

**Supervision:** William J. Chopik.

**Writing – original draft:** Lindsay S. Ackerman.

**Writing – review & editing:** Lindsay S. Ackerman, William J. Chopik.

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
