## [Decision Letter · Decision Letter 0]

12 Dec 2019

PONE-D-19-25949

Individual differences in personality predict the use and perceived effectiveness of essential oils

PLOS ONE

Dear Dr Chopik,

Thank you for submitting your manuscript to PLOS ONE. After careful consideration, we feel that it has merit but does not fully meet PLOS ONE’s publication criteria as it currently stands. Therefore, we invite you to submit a revised version of the manuscript that addresses the points raised during the review process.

Please address the issues raised by Reviewer 2.

We would appreciate receiving your revised manuscript by 12 January 2020. To enhance the reproducibility of your results, we recommend that if applicable you deposit your laboratory protocols in protocols.io, where a protocol can be assigned its own identifier (DOI) such that it can be cited independently in the future. For instructions see: http://journals.plos.org/plosone/s/submission-guidelines#loc-laboratory-protocols

We look forward to receiving your revised manuscript.

Kind regards,

Rosemary Frey

Academic Editor

PLOS ONE

Journal Requirements:

2. In the abstract, for clarify and considering the wide PLOS ONE readership, please update the sentence in the Abstract reading "We found that bullshit receptivity and religiosity..." to "We found that receptivity to pseudo-profound fabricated statements and religiosity..."

Reviewers' comments:

Reviewer's Responses to Questions

**Comments to the Author**

1. Is the manuscript technically sound, and do the data support the conclusions?

Reviewer #1: Yes

Reviewer #2: Partly

2. Has the statistical analysis been performed appropriately and rigorously? 

Reviewer #1: Yes

Reviewer #2: Yes

3. Have the authors made all data underlying the findings in their manuscript fully available?

Reviewer #1: Yes

Reviewer #2: Yes

4. Is the manuscript presented in an intelligible fashion and written in standard English?

Reviewer #1: Yes

Reviewer #2: Yes

5. Review Comments to the Author

Reviewer #1: Very good study. Explanation given was extraordinary. Research and publication ethics was satisfied. Given great introduction about the topic. Results were greatly explained in graph. Discussion part can improve better.

Reviewer #2: The analysis is straightforward enough, at least in terms of the regression methods and scales applied. (I've taught that for three decades.) Where the problem lies is in the author's lack of transparency and critical thinking as to how this issue was conceptualized in the first place. There is now a growing body of evidence, which the author conveniently ignores, in the medical and social sciences indicating that persons who are religious, or spiritual (not necessarily the same thing) in their orientation tend to have better medical outcomes, live longer, and are happier than those who are more skeptical and/or atheistic. As most practicing physicians are aware, the 'power of belief' in something, independent of its demonstrable scientific verifiability, sometimes sets in motion psycho-physiological processes (e.g. stress reduction) that can result in improved health. That we do not yet sufficiently understand how (what BSR scale proponents narrowly characterize as 'bullshit') works does not mean that the (seemingly irrational) rituals such as EO are entirely without value.

Unfortunately this article gives the reader the impression that it has a preexisting agenda to discredit complimentary and integrative medical approaches, particularly in its choice of the conceptually-biased BSR measure, and religiosity rather than a broader measure of spirituality. At minimum, the literature review and discussion sections need to acknowlege, not just the literatures of their preferred paradigm, but also of the paradigm one finds reflected e.g. in scientific articles in European Journal of Integrative Medicine.

Minor correction...

lines 48-49 should read

48 antioxidant, antibacterial, antifungal, antimicrobial, and antiplaque/antigingivitic properties, as

49 well //as// their performance as an effective insect repellant (4-10).

lines 59-60 should read

59 A separate but related question //as// to whether or not EOs provide marked health benefits is

60 whether people perceive benefits of EO use. Further, it is likely that some individuals are more

6. PLOS authors have the option to publish the peer review history of their article (what does this mean?). If published, this will include your full peer review and any attached files.

Reviewer #1: No

Reviewer #2: No

---

## [Author Response · Author response to Decision Letter 0]

12 Jan 2020

***Response to Reviewers***

We would like to thank the reviewers for their thoughtful comments on the manuscript. We very much appreciate the feedback and believe that the manuscript has improved significantly as a result of the suggestions in this round of review. We are happy to make any additional changes that the Editor feels is necessary.

The line numbers below refer to the clean (i.e., not tracked changes) version of the manuscript.

***Associate Editor***

1.) The Editor noted that we should adhere to PLOS ONE’s style and naming conventions. We have now edited the manuscript so that it is consistent with the templates linked to us by the Editor. 

2.) The Editor also recommended that we change “bullshit receptivity” to “receptivity to pseudo-profound fabricated statements” in the Abstract. We have now done so.

***Reviewer #1***

Reviewer #1 characterized our paper has containing a “very good study” and that the “explanation given was extraordinary.” They noted that research and publication efforts were satisfied, the introduction was great, and that the results were greatly explained. 

We thank the reviewer for their kind assessment of our paper. They noted that the Discussion could be improved and we have now done so according to some recommendations made by Reviewer #2.

***Reviewer #2***

1.) Reviewer #2 recommended that we integrate the literature on the health benefits of religiosity and spirituality. Indeed, there is an extensive literature on the health benefits of religious and/or spiritual individuals (and the mechanisms linking the two). We have now added these considerations to the Introduction (on pages 4, lns 58-74) and Discussion (on pages 14-15, lns 268-281). We also make a connection to how the use of essential oils reflects an engagement with religious traditions and preventative health behaviors more generally.

2.) Reviewer #2 also recommended that we discuss some limitations related to measurement, particularly of the BSR measure and our narrow measure of religiosity (instead of a broader measure of spirituality). We have now done so in the Discussion on pages 17-18 (lns 321-346). They also recommended that we cite literature from alternative paradigms, especially those published in a particular journal (e.g., the European Journal of Integrative Medicine). Although our methodological approach substantially differs from that used in studies examining the effectiveness of EOs, we have now added some of this literature to pg 3 (lns 45-49)

3.) Finally, the reviewer noted two minor corrections (inserting the word “as” into two sentences). We have now made these changes.

---

## [Decision Letter · Decision Letter 1]

14 Feb 2020

Individual differences in personality predict the use and perceived effectiveness of essential oils

PONE-D-19-25949R1

Dear Dr Chopik,

We are pleased to inform you that your manuscript has been judged scientifically suitable for publication and will be formally accepted for publication once it complies with all outstanding technical requirements.

With kind regards,

Rosemary Frey

Academic Editor

PLOS ONE

Additional Editor Comments (optional):

Reviewers' comments:

Reviewer's Responses to Questions

**Comments to the Author**

1. If the authors have adequately addressed your comments raised in a previous round of review and you feel that this manuscript is now acceptable for publication, you may indicate that here to bypass the “Comments to the Author” section, enter your conflict of interest statement in the “Confidential to Editor” section, and submit your "Accept" recommendation.

Reviewer #2: All comments have been addressed

2. Is the manuscript technically sound, and do the data support the conclusions?

Reviewer #2: Yes

3. Has the statistical analysis been performed appropriately and rigorously? 

Reviewer #2: Yes

4. Have the authors made all data underlying the findings in their manuscript fully available?

Reviewer #2: Yes

5. Is the manuscript presented in an intelligible fashion and written in standard English?

Reviewer #2: Yes

6. Review Comments to the Author

Reviewer #2: (No Response)

7. PLOS authors have the option to publish the peer review history of their article (what does this mean?). If published, this will include your full peer review and any attached files.

Reviewer #2: No

---

## [Editor Report · Acceptance letter]

28 Feb 2020

PONE-D-19-25949R1 

Individual differences in personality predict the use and perceived effectiveness of essential oils 

Dear Dr. Chopik:

I am pleased to inform you that your manuscript has been deemed suitable for publication in PLOS ONE. Congratulations! Your manuscript is now with our production department. 

With kind regards,

on behalf of

Dr. Rosemary Frey 

Academic Editor

PLOS ONE